# The Grammaticalization of the Discourse Marker *genre* in Swiss French

Delin Deng 

Department of Linguistics, University of Florida, Gainesville, FL 32611, USA; ddeng@ufl.edu

**Abstract:** By conducting an apparent-time analysis of the OFROM corpus collected in Francophone Switzerland, this study examined the use of *genre* as discourse marker in the speech of 306 French L1 speakers. First, we examined the effect of extralinguistic factors on the discursive use of *genre*. The logistic mixed-effects regression analysis results revealed that the emerging use of *genre* is indeed an ongoing change led by female speakers in Swiss French. This use was favored by monolinguals in Francophone Swiss. Second, we examined the vowel reduction of the DM *genre* in the corpus. Our results revealed that speakers who received only a high school education favor the vowel reduction in the DM *genre* the most. Given the high percentage of phonological reduction in the DM *genre*, we believe that the grammaticalization of this particle has reached its advanced stage in Swiss French. Compared to previous findings on the emerging use of *genre* in Hexagonal French, we suggested that the grammaticalization of the particle *genre* in Swiss French may be independent of that in Hexagonal French. The grammaticalization in Swiss French was much more advanced than in Hexagonal French. This study supplied comparable results on the grammaticalization of the same particle in two different Francophone countries.

**Keywords:** DM; grammaticalization; phonological reduction; variation and change



## 1. Introduction

Different scholars have documented the emerging discursive use of *genre* in native French (see, for example, Denison 2002; Mihatsch 2010; Secova 2011). Secova (2011), in particular, reported the use of *genre* as a discourse marker (DM) to be typical in youth language. Examples (1) and (2) illustrate the DM and non-DM use of *genre* in native speech.

(1)  le langage que j'ai des fois *genre* avec mes copines (OFROM_unine19-002)
     "the language I sometimes have like with my girlfriends"

(2)  enfin ouais plus ce *genre* de mots comme spoiler (OFROM_unine19-017)
     "well yes more this kind of words like spoiler"

As shown by these two examples, when used as a DM, a lexical item no longer belongs to the original category assigned to it. The original use of *genre* is a noun, as in example (2), while the DM *genre* in example (1) functions more like an adverb. The difference between this discursive and non-discursive use is that the removal of the former does not affect the semantic integrity of the sentence, while removing the latter does. This change from non-discursive use to discursive use of a lexical item often involves the process of grammaticalization.

Over recent decades, much research has been dedicated to describing the discursive functions of *genre*. Not much has been known regarding its use in a social context. Only marginal studies discussed this aspect by pointing out that this might be a feature of youth speech (see, for example, Secova 2011). However, we still do not know, for example, if the DM *genre* is fully grammaticalized in Swiss French or if it is still a change in progress. What is the supporting evidence for its grammaticalization if it is a change in progress? Is it a

change independent of that in Hexagonal French? What are the constraining social factors influencing this discursive use? etc.

Therefore, by conducting an apparent-time variationist analysis of the particle *genre* in Swiss French native speech, we hope to be able to answer, if not all, at least some of the questions raised here. The objective of the current work is to, from a quantitative point of view, on the one hand, discuss the grammaticalization of *genre* in Swiss French native speech by analyzing oral data taken from the online publicly accessible corpus and its correlation with social factors, such as age, gender, sociolinguistic situation, and socio-educational status of the speakers. On the other hand, it will also provide supporting evidence of a phonological reduction in the DM *genre* for its grammaticalization.

The structure of this article is as follows: Section 2, the relevant literature on grammaticalization and discursive functions of *genre* in French native speech is reviewed. Section 3, the methodology, including information on corpora, speakers, tokens and extralinguistic factors, as well as statistical analysis, is presented. Section 4, the results of the current study are presented and discussed in tables. Section 5, the article is concluded with a summary of the current work as well as future implications.

## 2. Theoretical Backgrounds

### 2.1. Process of Grammaticalization

Over the past century, ample research and debates have contributed to identifying the different stages of grammaticalization to explain how a lexical item becomes grammaticalized. It is believed that the notion of "grammaticalization" was first mentioned by Meillet (1912). He described grammaticalization as "the passage of an autonomous word into the role of grammatical element" (Meillet 1912, p. 131). Heine and Reh (1984, p. 15) argued that grammaticalization is "an evolution whereby linguistic units lose in semantic complexity, pragmatic significance, syntactic freedom, and phonetic substance". Traugott (1995, p. 1) defined grammaticalization as "the process whereby lexical material in highly constrained pragmatic and morphosyntactic contexts becomes grammatical, in other words, that lexical material in specifiable syntactic functions comes to participate in the structural texture of the language, especially its morphosyntactic constructions".

It is argued that, according to many, the process of grammaticalization is "unidirectional" (see, for example, Meillet 1912; Saxena 1995; Vincent 2001) in the sense that this process is irreversible (see, for example, Haspelmath 1999). Heine et al. (1991) believed that grammaticalization involves a combination of discrete stages and continuum. As pointed out by Kuryłowicz (1965, p. 52), "grammaticalization consists in the increase of the range of a morpheme advancing from a lexical to a grammatical or from a less grammatical to a more grammatical status". "Such a progression often involves a number of intermediate stages, making it difficult in many cases to maintain a neat distinction between lexical and grammatical elements" (Saxena 1995, p. 352).

Lehmann identified three parameters and processes of grammaticalization (Lehmann [1982] 1995, p. 164), as illustrated in Table 1:

**Table 1.** Lehmann [1982] (1995, p. 164) Correlation of grammaticalization process (excerpt).

| Parameter | Weak GR | Process | Strong GR |
|---|---|---|---|
| Scope | Item relates to constituent of arbitrary complexity | Condensation | item modifies word or stem |
| Bondedness | Item independently juxtaposed | Coalescence | item = affix, phonol, feature or carrier |
| Syntagmatic variability | Item can be shifted around freely | Fixation | item occupies fixed slot |

Within this framework, grammaticalization is viewed as a process of increased bondedness but decreased scope. Two aspects are central to the process of grammaticalization: phonological reduction and semantic bleaching. While some believe that phonological reduction and semantic bleaching often go hand in hand in the process of grammaticaliza-

tion, Haspelmath (1999, p. 1058) argued that semantic bleaching should be the cause of other processes of grammaticalization in that "lexical items that fulfill a frequent discourse function will then increase in frequency because they are very often useful". He further pointed out that "increased frequency also means increased predictability, and the more predictable an item is, the less phonologically salient it needs to be" (Haspelmath 1999, p. 1058). This is mainly because when it becomes highly frequent, the probability of misunderstanding the lexical item with phonological reduction decreases accordingly.

Later, Lehmann (2015) further identified different phases of grammaticalization.

As illustrated in Figure 1, the morpho-phonemic change only occurs at a later phase of grammaticalization. As pointed out by Haspelmath (1999), it is at this stage of development that the lexical item in question has reached a high frequency in use and that the phonological reduction does not affect the listener's comprehension. That is to say, if we can identify any phonological reduction in the grammaticalization of a lexical item, the process of grammaticalization for this lexical item has probably reached a relatively advanced stage.

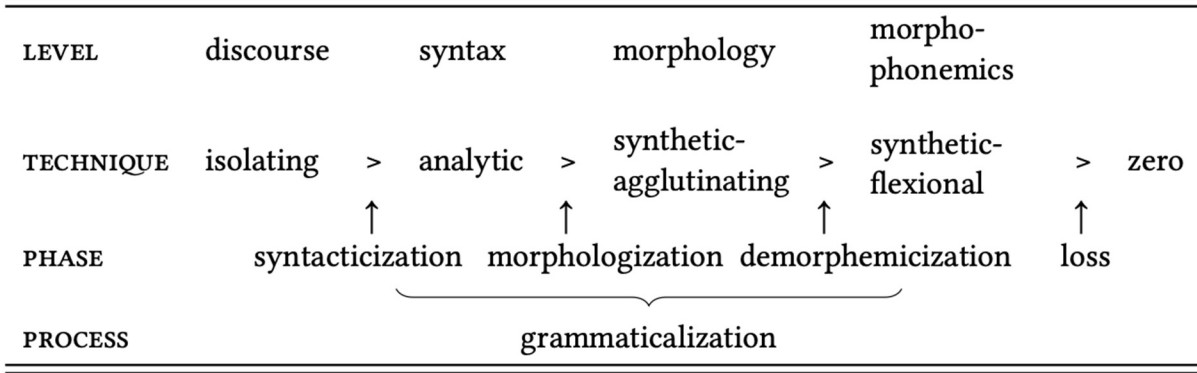

**Figure 1.** Phases of grammaticalization (Lehmann 2015, p. 15).

This discussion of grammaticalization becomes even more relevant when talking about DMs. As defined by Hansen (1998), DMs are "linguistic items which fulfill a non-propositional, meta-discursive (primarily connective) function, and whose scope is inherently variable, such that it may comprise both sub-sentential and supra-sentential units". It is usually the case that when a lexical item is used as a DM, it no longer belongs to the original grammatical category assigned to it but is semantically independent of the sentence. Removing a DM from the carrying sentence does not affect the grammaticality of the sentence. The impact is only at the pragmatic level. As pointed out by Fraser (1988, p. 22), "the absence of the DM does not render a sentence ungrammatical and/or unintelligible. It does, however, remove a powerful clue about what commitment the speaker makes regarding the relationship between the current utterance and the prior discourse".

### 2.2. Genre Used as a DM in Native Speech

When did *genre* start to be used as a DM? What are the pragmatic functions it fulfills when used as a DM? Mihatsch (2010) pointed out that the form of *genre de* began to be used as an approximator as early as the 15th century. Nevertheless, it is generally believed that *genre*, the shortened form of *genre de*, is a newly emerged and very frequent DM in European French. As confirmed by Secova (2011), it is difficult to establish the diachronic development of the appearance of *genre* as a particle since it is not documented in any dictionaries and only exists in some spoken corpora. Denison (2002) assumed that the discursive use of *genre* probably developed from its qualifying particle use by semantic bleaching and blurring of scope boundaries. It is noticed that the English *like*, Canadian French *comme*, and European French *genre* do share some pragmatic similarities. Mihatsch (2010) identified three functions of *genre* in European French: adaptor, quotative use, and rounder. Secova (2011) differentiated between approximation; exemplification/paraphrasing; reporting

speech, thought and attitude; and expressing irony. She also pointed out that at the epistemic level *genre* seems to be used as a hedge while at the syntax-semantic level as a marker of focus. We will illustrate with examples below based on Secova's classification. The same functions could be identified in both Hexagonal and Swiss French.

### 2.2.1. Approximation

It is the primary function of *genre* as a particle to indicate the approximation or inexactitude, functioning both as adaptors and rounders.

(3)  ils ont fait du bruit franchement … il était un truc *genre* trois—quatre heures du matin (Secova 2011, p. 100)
"they made noises frankly … it was a thing like three-four o'clock in the morning"

It is clear here that *genre* modifies the numeral value and overlaps the rounder function of *comme* in example (3). *Genre* could be paraphrased by *environ* or *à peu près*. By using *genre*, the speaker indicates the vagueness of what was said by attenuating the force of the speech act.

### 2.2.2. Exemplification

In this case, *genre* is often used in the utterance-initial position and connects the previous utterance with the following utterance. *Genre* either provides an example or a justification for the previous utterance.

(4)  elle parle toute seule … *genre* t'as vu quand elle était sur le canapé? (Secova 2011, p. 104)
"she talks all alone … like you saw when she was on the couch?"

### 2.2.3. Speech Reporting Speech

The quotative *genre* can usually be substituted by the verb *say*. It introduces direct speech without being too serious about what is quoted. In this sense, the use of *genre* reduces the formalness of the speech.

(5)  moi j'ai bien aimé … franchement … mais Patrick il était là *genre* "ouais j'aime pas la chanteuse" (Secova 2011, p. 106)
me I loved a lot … frankly … but Patrick he was there like "yeah I don't like the singer"

### 2.2.4. Expressing Irony

It seems that this function is unique to *genre*. The ironic *genre* often offers a quotation or explanation with an ironic tone and could probably be paraphrased by the French phrase *soi-disant* "so-called". By using *genre*, the speaker reduces the credibility of the utterance.

(6)  tu sais à quelle heure elle nous remplace son cours *genre* pour pas nous déranger? à huit heures Samedi!!! (Fleischman and Yaguello 2004, p. 137)
"you know at what time she replaces us with her lesson like not to disturb us? at eight o'clock Saturday!!!"

As shown in example (6), the fact that the replacement of the class is at eight o'clock on a Saturday morning is an act of disturbing others. After the ironic *genre*, the speaker used the expression "not to disturb us," where he actually meant the opposite. The *genre* here constitutes a drastic contrast between what is said and what is intended by the speaker. In some way, *genre* reinforces the force of the hidden meaning.

Despite the depth of research on discursive functions of *genre* and the historical development of its use as a DM, to the best of our knowledge, no research has documented the phonological reduction of this particle when used as a DM. However, as illustrated in the literature, phonological reduction is strong evidence for the grammaticalization of lexical items. Previous studies have mainly focused on describing its discursive functions. There has not been any attempt to explore how social factors, such as the age, gender, or socio-economic status of the speakers, could impact its use as a DM. Previous studies on language variation and change have demonstrated that social factors could be important

indicators for the development of linguistic variables in a speech community (see, for example, Labov 2001). A variationist study of *genre* at binary classification of DM *genre* and non-DM *genre* could shed more light on this topic.

Therefore, in this work, by conducting logistic mixed-effects regression analysis in R using Rbrul (Johnson 2009; R Core Team 2021) on *genre* in OFROM corpus (le corpus Oral de Français de Suisse Romande, (Avanzi et al. 2012–2020), www.unine.ch/ofrom (accessed on 11 November 2022), we tried to answer the following research questions: Is this discursive use of *genre* an ongoing change in Swiss French in apparent time? What are the social factors that impact this discursive use? Is there any other evidence, such as phonological reduction, that supports the claim that *genre* is undergoing the process of grammaticalization?

## 3. Methodology

### 3.1. Corpus

The corpus used in this study is OFROM. OFROM is a text-sound-aligned publicly accessible online corpus consisting of some sociolinguistic interviews conducted with speakers of French in Switzerland. All the interviews were orthographically transcribed in Praat (Boersma and Weenink 2021). The corpus was initiated in 2012 and is still under construction. As of 2020, the corpus contains 342 speakers, totaling 64 h of recording and 1,005,000 words. For the current study, we only used data collected in seven Francophone cantons, where French is the official language, in Switzerland. Therefore, only 306 speakers were included in our final analysis. These seven cantons are Neuchâtel, Fribourg, Valais, Vaud, Jura, Bern, and Geneva. The geographic distribution of these seven cantons is shown in Figure 2.

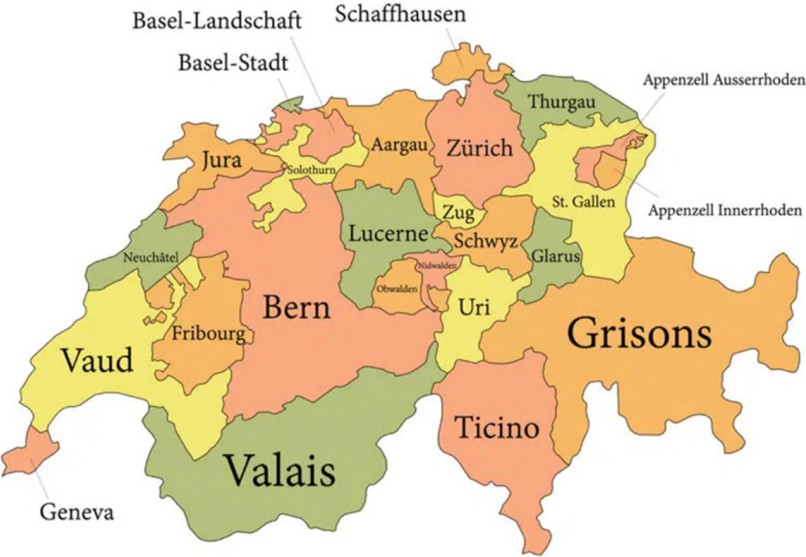

**Figure 2.** Geographic distributions of cantons in Switzerland (map adapted from: https://holidaystoswitzerland.com/regions-and-cantons-of-switzerland/ (accessed on 11 November 2022).

As shown in Figure 2, the seven cantons are located in the western part of Switzerland. Based on their sociolinguistic situations, they can be further divided into two groups: monolingual cantons and bilingual cantons. Jura, Neuchâtel, Vaud, and Geneva are monolingual Francophone cantons, while Bern, Fribourg, and Valais are bilingual cantons where French and German are spoken.

### 3.2. Speakers

Since OFROM contains both speakers of L1 and L2 French, only native speakers of French were included in our analysis. Therefore, the speakers in the current study are all native speakers of Swiss French living in Switzerland at the time of the interview within

the seven cantons mentioned above. All the speakers were born between 1932 and 2001. As age is a factor in the current study, speakers whose year of birth was missing from their profile were excluded from the final analysis. Thus, 306 speakers were included in our final analysis. The distribution of speakers in each canton is presented in Figure 3.

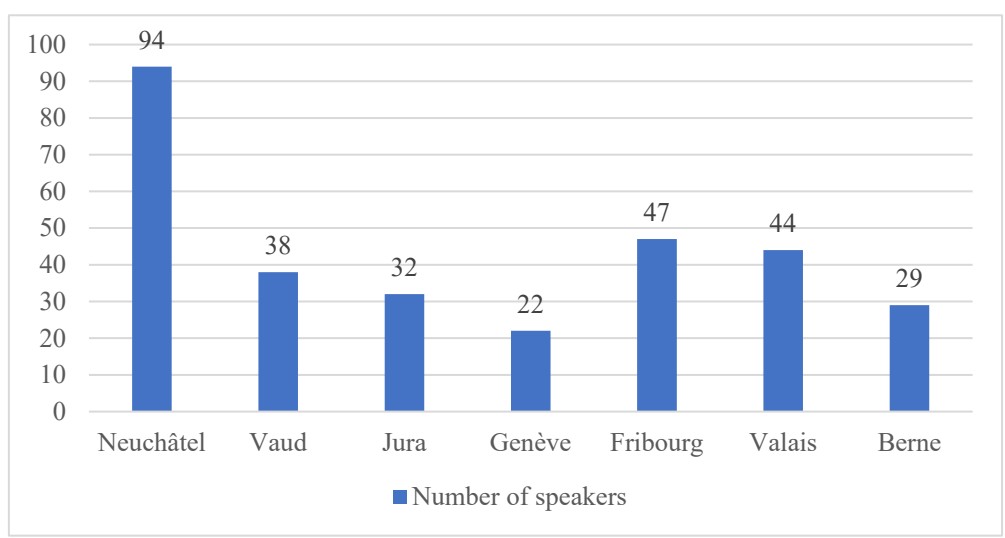

**Figure 3.** Distribution of speakers in seven cantons.

*3.3. Tokens*

In total, 587 occurrences of *genre* were identified in the OFROM corpus. Each token was first coded for its discursive status by the principal investigator. More precisely, every token is coded either as DM *genre* or non-DM *genre.* As shown in Example 1 and Example 2, the *genre* used as a noun and that can be substituted by *sorte* or *type* was coded as a non-DM *genre*, while *genre* in all other uses was coded as the DM *genre*. As shown in Figure 4, 75% of uses of *genre* were as a DM, and 25% were as a non-DM. According to our data, *genre* is mainly used as a DM in Swiss French, which suggests the grammaticalization of the particle.

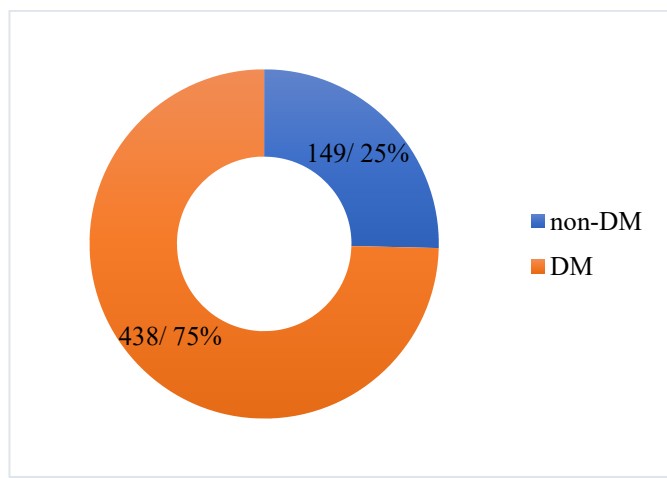

**Figure 4.** Distribution of the DM genre and non-DM genre in OFROM.

After being coded for their discursive status, all the tokens of *genre* were coded for their vowel quality. By listening to all the tokens and inspecting the spectrogram, we mainly coded for two realizations: *genre* without phonological reduction, realized as [ʒãʁ] and *genre* with phonological reduction, realized as [ʒœʁ]. The difference is shown in Figure 5.

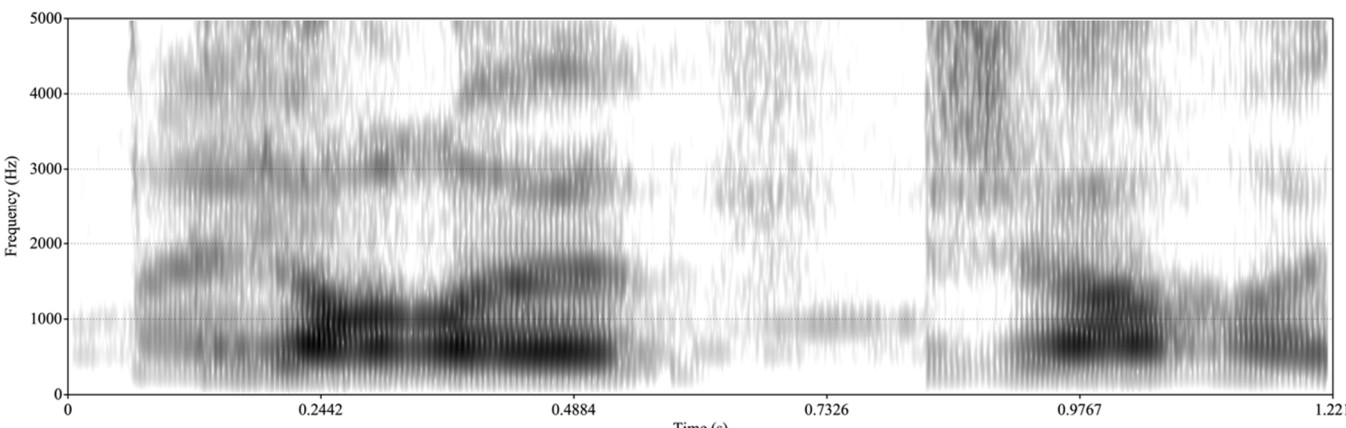

**Figure 5.** Spectrogram for *genre* [ʒɑ̃ʁ] (left) without vowel reduction and *genre* [ʒœʁ] (right) with vowel reduction.

Figure 6 shows the distribution of *genre* realized with phonological reduction and without phonological reduction in both DM and non-DM uses. As shown in Figure 6, in its original nominal status, only 32 out of 149 uses of *genre* (21.48%) were realized with phonological reduction. However, when *genre* was used as a DM, 187 out of 438 uses of *genre* (42.69%) were realized with phonological reduction. This indicates that when used as a DM, *genre* is more susceptible to phonological erosion. This observation corresponds to the grammaticalization process of particles in general, as shown in the literature.

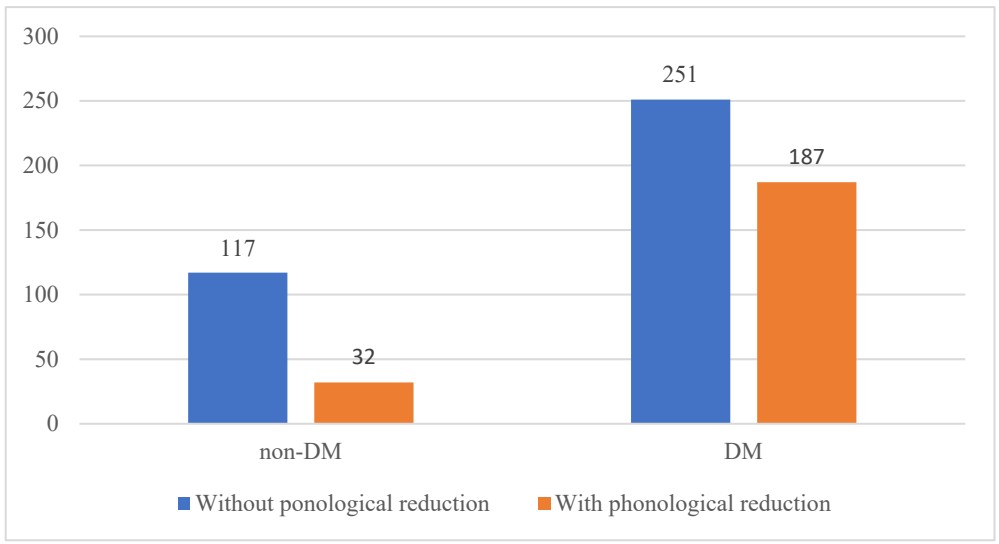

**Figure 6.** Distribution of phonological reduction in both DM and non-DM use.

*3.4. Extralinguistic Factors*

For extralinguistic factors, we looked at the age, gender, sociolinguistic situation, and social-educational status (SES) of the speakers, as shown in Table 2.

**Table 2.** Extralinguistic factors to be examined.

| Predictors | Levels |
| :---: | :---: |
| Gender | female |
|  | male |
| Sociolinguistic situation | monolingual |
|  | non-monolingual |
| Socio-educational status (SES) | level 1 |
|  | level 2 |
|  | level 3 |
|  | level 4 |
| Age | continuous |

For the "age" factor, we use the year of birth provided by the personal information in the corpus. For this factor, we aimed to determine, on the one hand, if the discursive use of the DM *genre* is an ongoing change in Swiss French, and on the other hand, if there is any age-grading effect on the phonological reduction when used as a DM. In either case, we should expect the "age" factor to be significant.

For the "gender" factor, we mainly want to determine whether the use of *genre* is particularly favored by any gender group. As Blondeau and Deng (forthcoming) demonstrated, in Hexagonal French, the discursive use of *genre* is favored mainly by male speakers and, therefore, is socially marked. We aimed to assess whether the same applies to the discursive use of *genre* in Swiss French.

Regarding the "sociolinguistic situation" factor, as mentioned earlier, four of the seven cantons are monolingual, and the other three are bilingual. For this factor, we mainly distinguish between two levels: monolingual and bilingual. We aimed to determine whether monolingual status facilitates the discursive use of *genre*. If the discursive use of *genre* is mainly associated with bilingual speakers, language contact might play a role. In contrast, if this use is more associated with monolingual speakers, the emergence of the discursive use of *genre* may not be the consequence of contact with another language.

As for the "socio-educational status (SES)" factor, four levels are distinguished in the corpus: Level 1: compulsory education with technical learning; Level 2: compulsory education with office learning; Level 3: high school education; Level 4: university education. Since socio-educational status is closely related to the social-economic status of the speakers, for this factor, we aimed to determine whether the discursive use of *genre* is particularly favored by speakers from any social class.

*3.5. Statistical Analysis*

For this article, we performed the logistic mixed-effects regression analysis in R using Rbrul (Johnson 2009; R Core Team 2021). The model distinguished the following levels for statistical significance: $p > 0.1$, not significant; $0.05 < p < 0.1$, marginally significant; $p < 0.05$, significant; $p < 0.01$, very significant; $p < 0.001$, highly significant. For the results, the model provided one *p*-value for each predictor (the independent variable) to indicate whether this predictor is statistically significant for predicting the dependent variable. It also provided the factor weight and log odds for each level of the predictor to indicate which levels favor/disfavor the chosen variable.

For our analysis, we first examined the dependent variable, which is *genre*, at binary classifications of the DM *genre* vs. non-DM *genre*. Second, we looked at the dependent variable, which is the DM *genre* at binary classifications of the DM *genre* with phonological reduction vs. DM *genre* without phonological reduction. For both, the fixed independent variables are extralinguistic factors presented above. All fixed factors except for age are categorical. As we used the birth year of the speakers for the age, the age factor is thus continuous. To include the mixed effects, we used speakers as a random variable. For the modeling, we performed the one-level test. Since the "participants" factor was treated as a

random variable, for the following section, we only provide the results for the fixed factors for further discussion.

## 4. Results

### 4.1. Discursive Use of genre

Overall, the logistic mixed-effects regression analysis indicated that the discursive use of *genre* is constrained by different social factors. As shown in Table 3, the gender, age, and sociolinguistic situation of the speakers are proven to be statistically significant to the discursive use of *genre* in Swiss French. In contrast, the socio-educational status of the speakers is not statistically significant to this use.

**Table 3.** Correlation between the discursive use of genre and extralinguistic factors.

| | | DM *genre*/Non-DM *genre* | |
|---|---|---|---|
| Input prob. | | 0.746 | |
| Total number | | 587 | |
| Log. likelihood | | −267.83 | |
| | F. w | % | N |
| **Gender** | | *p* = **0.000864** | |
| female | 0.668 | 82.9 | 398 |
| male | 0.332 | 57.1 | 189 |
| **Age** | | *p* = **0.00136** | |
| continuous | log odds | | |
| +1 | 0.047 | | |
| **Sociolinguistic situation** | | *p* = **0.0688** | |
| monolingual | 0.603 | 78.3 | 470 |
| non-monolingual | 0.397 | 59.8 | 117 |
| **SES** | | Not significant | |
| Level 1 | [0.596] | 62.5 | 8 |
| Level 2 | [0.511] | 78.9 | 76 |
| Level 3 | [0.502] | 82.2 | 129 |
| Level 4 | [0.391] | 71.4 | 374 |
| **Speakers** | | Random | |

As presented in Table 3, the "age" factor is statistically very significant to the discursive use of *genre* ($p < 0.01$; log odds: 0.047). As shown by the results, the larger the value of the "age" factor, the more likely a speaker will use *genre* as a DM. Since we use the year of birth of the speaker for this factor, as the value of the factor increases, the actual age of the speaker decreases. Thus, the larger the number, the younger the speaker. Therefore, the discursive use of *genre* is age-graded in apparent time with a strong association with younger speakers in the speech community. This proves that the discursive use of *genre* is indeed an ongoing change in Swiss French.

Meanwhile, the "gender" factor is proven to be highly significant to the discursive use of *genre* ($p < 0.001$; f.w.: female: 0.668; male: 0.332). Female speakers favor the discursive use of *genre*, while male speakers disfavor this use. Combined with the results for the "age" factor, our results indicate that the discursive use of *genre* is an ongoing change led by female speakers.

Regarding the "sociolinguistic situation" factor, it is only marginally significant to the discursive use of *genre* ($0.05 < p < 0.1$; f.w.: monolingual: 0.603; bilingual: 0.379). The monolinguals are more likely to use *genre* as a DM, while bilinguals are less likely to do so. This suggests that the emergence of the discursive use of *genre* in Swiss French is likely not due to language contact with other languages in the speech community. This, however, partially corroborates previous findings on *genre* in other French varieties that it is not a

calque of its English equivalent *be like* but rather an independent internal development that leads to the grammaticalization of the particle (see, for example, Cheshire and Secova 2018).

### 4.2. Phonological Reduction of the DM genre

As shown in Table 4, the only social factor that is statistically significant to the vowel reduction in the DM *genre* is the socio-educational status of the speakers. Gender, age, and sociolinguistic status are not significant to the vowel reduction in the DM *genre*. Our results indicated that speakers from level 3 SES are the group that tends to reduce the vowel in the DM *genre*. In total, 75.5% of the DM uses of *genre* produced by this group are found to be with vowel reduction. All other three groups disfavor vowel reduction. This suggests that the phonological reduction of *genre*, when used as a DM, is highly socially marked. By percentage, we can see that speakers with high SES are more likely to reduce the vowel in the DM *genre*, while speakers with lower SES are less likely to have vowel reduction in their DM *genre*.

**Table 4.** Correlation between the phonological reduction of the DM genre and social groups.

| | With Vowel Reduction/Without Vowel Reduction | | |
|---|---|---|---|
| Input prob. | 0.427 | | |
| Total number | 438 | | |
| Log. likelihood | −222.966 | | |
| | F. w | % | N |
| **SES** | ***p* = 0.0426** | | |
| Level 3 | 0.827 | 75.5 | 106 |
| Level 4 | 0.48 | 36.0 | 267 |
| Level 1 | 0.365 | 20.0 | 5 |
| Level 2 | 0.283 | 16.7 | 60 |
| **Gender** | **Not significant** | | |
| female | 0.525 | 45.2 | 330 |
| male | 0.475 | 35.2 | 108 |
| **Age** | **Not significant** | | |
| continuous | log odds | | |
| +1 | 0.014 | | |
| **Sociolinguistic situation** | **Not significant** | | |
| monolingual | 0.601 | 45.1 | 368 |
| non-monolingual | 0.399 | 30.0 | 70 |
| **Speakers** | Random | | |

## 5. Discussion: Emergence of the DM *genre* in Swiss French

As presented earlier in this article, the OFROM corpus we used in the current work was initiated in 2012, while the one Blondeau and Deng (forthcoming) used in their study of Hexagonal French was ESLO 2 was initiated in 2008. Based on the time of corpus construction, these two corpora offered comparable oral data on two French varieties in different countries. However, when it comes to the use of *genre* in these two corpora, we notice that the percentage of this discursive use in these two corpora differs significantly. As shown by our results, 75% of uses of *genre* were as a DM in Swiss French. Blondeau and Deng (forthcoming) reported that only 47.3% of uses of *genre* were as a DM in Hexagonal French. That is to say, the discursive development of *genre* in these two regions is not at the same pace or, at least, is not at the same developmental stage, even though in both studies, the discursive use of *genre* was reported to be an ongoing change in French in both countries. The discursive use of *genre* is much more advanced in its development in Swiss French.

At the same time, it is also interesting to see the difference in gender impact on the discursive use of *genre*. In Swiss French, it is women who led the change, while in Hexagonal French, at a comparable time, this discursive use had already spread to both genders. It is intriguing that in Blondeau and Deng (forthcoming), it is reported that in the earlier corpus ESLO 1, the use of *genre* as a DM is more associated with male speakers. Compared with our results here on gender effect, it is curious to see how the discursive use of the same particle, though an ongoing change in both countries, could be a change led by different gender groups. The original status of the DM *genre* seems to be very different in France and Switzerland. The DM *genre* in Hexagonal French is more strongly correlated with a certain gender than in Swiss French at the initial stage, while this marked status gradually disappeared over 40 years. In contrast, in Swiss French, women led the change to its advanced stage. It suggests that the DM *genre* in Swiss French and Hexagonal French are two independent processes at different stages of grammaticalization.

As demonstrated in the literature, when a particle starts to undergo the process of grammaticalization, it also begins to lose its semantic complexity. Thus, semantic bleaching is often attested at the advanced stage of development. This semantic bleaching then leads to phonological reduction since it necessarily increases the predictability of the lexical item and, in consequence, reduces the necessity of its phonological saliency. Based on our results and previous literature, it is reasonable to expect more phonological reduction in Swiss French than in Hexagonal French since the grammaticalization of *genre* is much more advanced in Swiss French. This also justifies the examination of the phonological reduction of *genre* when used as a DM in this study.

As shown in Figure 6, 42.69% of the DM uses of *genre* are realized with phonological reduction. This confirms our hypothesis that phonological reduction should be expected at the advanced stage. What is even more intriguing about phonological reduction is that in our results, it is noticed that only speakers from level 3 SES favor this variant, while speakers from other SES groups do not favor this variant and realize the DM *genre* in its unreduced form. We cannot help but ask what is special about this speaker group. Why do they behave differently from speakers from other SES groups?

Coming back to this grouping, we notice that level 3 SES speakers are mainly speakers who have completed high school education. This particularity of their language use seems more related to the theory of the "critical age". As proposed by Labov (2001), the peak of the "critical age" is at 17 years old, when high schoolers intentionally calibrate their use of language according to their peers, a point also reckoned with by Eckert (1988, 1997). The speakers from level 3 SES are the ones that are mostly aware of the speech of their peers, while speakers from level 1 and level 2 SES may not have the chance to interact actively with their peers during the period of peer awareness since they never got that degree before they started working. As for level 4 SES speakers, as they continue to receive a university education and become more likely to use standard forms, they may intentionally use more standard linguistic forms in their speech. This result is particularly interesting because it were not attested in Hexagonal French in the literature. It supplies further evidence that the grammaticalization of the DM *genre* might be an independent process from the one in Hexagonal French, though the same use could be attested in French around the same period. Given the different developmental stages and phonological reduction degree, we have reason to believe that the two grammaticalization processes may not have a mutual influence.

Another point that needs further discussion in our results is the influence of the sociolinguistic status of the speakers on the discursive use of *genre* in Swiss French. As mentioned earlier in the article, due to the complexity of linguistic situations in Switzerland, the grammaticalization of a particle could, by all means, be the consequence of the language contact between different languages spoken in the region(s). Even though the overall percentage of *genre* used as a DM is high in the speech community, it is higher in the monolingual regions, where French is the only official language, than in the bilingual regions, where French and German are both the official languages. This gives us

reason to believe that the grammaticalization of *genre* in Swiss French is not a consequence of language contact in the Francophone regions since if it is language-contact-induced, we should see a higher percentage of discursive use in the bilingual region than in the monolingual region.

## 6. Conclusions

By conducting an apparent-time analysis of corpus collected in Francophone Switzerland in 2012, this article examines the variable use of the French discourse marker *genre* in the speech of 306 native speakers of Swiss French. The objective of the current study was to find further evidence of the grammaticalization of *genre* in Swiss French and discuss whether it is an independent process of grammaticalization from that in Hexagonal French documented in the literature.

The logistic mixed-effects regression analysis results revealed that the use of *genre* as a DM is indeed an ongoing change led by female speakers in Swiss French. Our analysis also proved that the phonological reduction of this particle is mainly associated with speakers with high school education. We posited that this might be related to the critical period when speakers are more sensitive to the speech of their peers. It is not the result of language contact in that speakers in French monolingual regions are more likely to use it as a DM than speakers in bilingual Francophone regions in Switzerland. Our results also demonstrated that it is at a more advanced stage of grammaticalization compared to the grammaticalization of this particle in Hexagonal French. We suggested that the grammaticalization of *genre* in Swiss French is independent of that in Hexagonal French and that it is not due to mutual influence. Our results also indicated that the DM *genre* in Swiss French has a higher percentage of phonological reduction than in Hexagonal French. This also confirms our earlier claim that the grammaticalization of *genre* in Swiss French is more advanced than in Hexagonal French.

Overall, our results supplied further evidence of the grammaticalization of the French discourse marker *genre* in Swiss French. The contribution of the current study is twofold. On the one hand, it supplies comparable results on the grammaticalization of the same particle in two different French varieties and thus provides further evidence for the independence of the grammaticalization of a particle in different varieties of a language; on the other hand, our data also shed new light on the language variation and change of discourse markers in French and thus provide new insights into the development of the same particle in different regions being conditioned by different social factors. While the ongoing change of *genre* is led by female speakers in Swiss French, it is an ongoing change first led by male speakers, then quickly spread to both gender groups in Hexagonal French. The age-grading effect on this particle is observable in Swiss French, while it is absent from Hexagonal French.

However, some questions remain untouched by the current study. For future studies, several venues could be taken. First, as seen earlier in the literature, the DM *genre* has different discursive functions in French native speech. Very often, the discursive functions of a particle do not always stay the same in its discursive development. Some functions might emerge at a certain point in history and enter the competition with other existing discursive functions of the particle, while other functions might become obsolete and gradually disappear. In a future study, it would also be relevant to examine whether the discursive functions of *genre* change over 40 years.

Second, since we only have a general idea about the origin of DM *genre*, it is preferrable to be verified in a more detailed diachronic corpus study. That is to say, it would be ideal to look into historical data to track the development of this particle diachronically. This would also provide a further comparison to its counterpart in Hexagonal French. If it is already more advanced than its counterpart in Hexagonal French, it is also possible that this discursive use began earlier in Swiss French. Looking at the historical data would also help us to understand at the initial stage how this particle enters the process of grammaticalization.

**Funding:** This research received no external funding.

**Institutional Review Board Statement:** The study did not require ethical approval. Ethical review and approval were waived for this study as it involved no experimenting with humans. The presented analyses were carried out on an already existing corpus of recordings at the Université de Neuchâtel that is publicly accessible online. The practice included informed consent from the recorded participants and reassurance by the institution that the recordings were anonymous and strictly confidential, meant to be used only for research purposes by the institution.

**Informed Consent Statement:** Informed consent was obtained from all subjects involved in the study.

**Data Availability Statement:** Not applicable.

**Conflicts of Interest:** The author declares no conflict of interest.

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
