# Peer review of "The Grammaticalization of the Discourse Marker genre in Swiss French"

_languages, doi:10.3390/languages8010028_

Round 1

Reviewer 1 Report

The paper is very interesting, well articulated and well written. The results are interesting, but it kind of lacks some discussion regarding the sociolinguistical aspects of the grammatical change observed. The PDF attached contains a list of comments, some references suggestions and other minor proof-reading checks.

Overall, we recommend to accept the paper, after minor revisions.

Author Response

Thank you very much for your comments and suggestions. They are very helpful in revising my article. I have made corresponding changes based on these comments and suggestions.

Reviewer 2 Report

The article is well written and clearly states the aims. The topic is interesting and opens up new research paths into the comparison of grammaticalization processes in the different varieties of contemporary French. I offer below some comments to improve the scientific presentation.

-      Paragraph 2.2: It would be interesting to explicit at this point which theory the author chooses for his work or if he/she prefers to present his/her own categorization.

-        Lines 191-195: It is not clear that the values described apply rather to Hexagonal French or Swiss French (or both). It seems the author defends that genre has the same values in both varieties, but it does not appear clearly in the article.

-        Lines 197-206: Again, explicit the description of the different values the author adheres to.

-        Lines 208-214: I suggest to say here if the different uses of the DM are going to be taken in account in the rest of the article or just the difference between DM and non DM uses.

-        Paragraph 3.2 : precise how many speakers in total after excluding some problematic ones.

-        Paragraph 3.3: Again, explicit if you are taking in account the different values of the DM presented, or just the dichotomy MD vs. non-MD.

-        Figure 6: it would be interesting to cross these results with the “age” variable.

-        Line 349: Could the author suggest an age range?

-        Lines 347-396:It would be interesting to cross this information with the diatopic variations within Swiss French. Could this differences depend also on the region?

-        Lines 423-437: Do you refer to speakers that are a certain age AND go to high school, or to speakers who only studied until high school but they could be 14, 20, 38 or 60? It would be good to add this precision also in the conclusion.

Other minor errors/comments:

-        Line 160: put “environ” instead of “environs”

-        Line 172: put “say” in italics

-        Line 266: put both “genre” in italics

-        Line 333: put “genre” in italics

-        Line 340: put “genre” in italics

-        Line 380: add year of publication after “Blondeau and Deng”

Author Response

(The authors gave the same response as above.)
